# Mucin-Producing Lobular Breast Carcinoma Metastasis to an Ovarian Fibroma: Histopathological and Immunohistochemical Analysis of a Rare Case and Literature Review

**DOI:** 10.3390/diagnostics14090953

**Published:** 2024-04-30

**Authors:** Alexandra Corina Faur, Camelia Vidiţa Gurban, Ecaterina Dăescu, Răzvan Vlad Tîrziu, Daniela Cornelia Lazăr, Laura Andreea Ghenciu

**Affiliations:** 1Department I, Discipline of Anatomy and Embryology, “Victor Babes” University of Medicine and Pharmacy, 300041 Timisoara, Romania; faur.alexandra@umft.ro (A.C.F.); daescu.ecaterina@umft.ro (E.D.); 2Department IV Biochemistry and Pharmacology, Discipline of Biochemistry, “Victor Babes” University of Medicine and Pharmacy, 300041 Timisoara, Romania; 3Department IX, Surgery I, Discipline of Surgical Semiology I and Thoracic Surgery, “Victor Babes” University of Medicine and Pharmacy, 300041 Timisoara, Romania; razvantarziu@umft.ro; 4Department V Internal Medicine I, Discipline of Internal Medicine IV, “Victor Babes” University of Medicine and Pharmacy, 300041 Timisoara, Romania; lazar_daniela@yahoo.com; 5Department III, Discipline of Pathophysiology, “Victor Babes” University of Medicine and Pharmacy, 300041 Timisoara, Romania; bolintineanu.laura@umft.ro

**Keywords:** breast cancer, lobular carcinoma, ovarian metastases, ovarian metastases from non-gynecologic primary sites, tumor-to-tumor metastases

## Abstract

Breast cancer stands as the primary cause of cancer-related mortality among women worldwide, often presenting with distant metastases upon diagnosis. Ovarian metastases originating from breast cancer represent a range of 3–30% of all ovarian neoplasms. Case Report: Herein, we present the histopathological, histochemical, and immunohistochemical findings of a rare case involving mucin-producing lobular breast carcinoma metastasizing to an ovarian fibroma in an 82-year-old female previously diagnosed with lobular breast carcinoma. Histopathological examination of the excised tissues revealed a biphasic neoplasm characterized by tumor cells expressing AE-1/AE-3 cytokeratin, mammaglobin, GCDFP-15, inhibin, and calretinin. Positive mucin staining was observed using histochemical techniques, and reticulin fibers were demonstrated using the Gordon–Sweets technique. A final diagnosis of mucin-producing lobular breast carcinoma metastatic to a benign ovarian fibroma was rendered. Conclusion: The occurrence of metastatic breast carcinoma overlaid on an ovarian tumor represents a rare and diagnostically challenging scenario.

## 1. Introduction

Metastases from remote primaries to the ovary are an uncommon condition. Approximately 3–30% of ovarian tumors are metastases and originate mainly from the gastrointestinal tract [1]. Metastases from tumors of the breast, pancreas, endometrium, bladder, lungs, and kidney are also reported in a small number of cases [1,2,3,4,5,6,7,8,9,10,11]. Most ovarian metastases are from extragenital organs, with 6.7% being breast carcinomas [4]. In rare cases, reports of tumor metastasis in an ovarian tumor or concurrent tumors with ovarian involvement are to be found [12,13,14,15,16,17,18,19]. Accounting for 5–15% of all breast cancers, lobular breast carcinomas (ILCs) are the second most common histological type. Invasive breast carcinoma sites of metastases include the bones, brain, liver, lungs, and lymph nodes [6,20,21,22]. Unusual locations like the gastrointestinal tract, uterus, ovary, peritoneum, retroperitoneum, bladder, spleen, and cervix can be involved in the metastatic spread of ILC [23,24,25,26,27,28,29]. Histologically, ILC has distinct morphologic features represented by round tumor cells lacking cohesion arranged in files, cords, and single cells that can also form targetoid patterns. The stromal response in ILC is minimal, and necrosis is uncommon. Morphological variants of ILC are classic lobular carcinoma, solid, alveolar, tubulolobular, pleomorphic, histocytoid, apocrine, signet ring, and mixed [30]. Also, variants of ILC with mucin production have been described [30,31,32,33].

Fibromas constitute common benign stromal ovarian sex-cord tumors, accounting for 1–4% of all ovarian neoplasms [34,35,36]. Ovarian fibromas typically present as unilateral, well-circumscribed solid masses. Histologically, these tumors consist of spindle cells with oval nuclei and scant cytoplasm. Due to their morphological resemblance, a differential diagnosis must include lesions such as thecomas, diffuse adult granulosa-cell tumors, massive ovarian edema, and fibrosarcomas. Clinically, consideration should be given to Meigs syndrome, characterized by ascites and pleural effusions in the presence of a benign ovarian mass, such as fibromas or fibroid tumors [37,38,39].

We hereby present a rare case of lobular breast carcinoma metastasizing to an ovarian fibroma.

## 2. Materials and Methods—Case Report

### 2.1. Presentation and Clinical Characteristics

An 82-year-old female with a previous diagnosis of lobular breast carcinoma presented to the Surgery Department of City Hospital Timisoara complaining of abdominal distension and pain. A right-side ovarian tumor was investigated ultrasonographically, and the clinician made a right ovarian tumor diagnosis. The laboratory data showed elevated values of cancer antigen 125 (CA-125) and human epididymis protein (HE-4) with a 95% ROMA score (risk of ovarian malignancy algorithm). Because of the high likelihood of malignancy, the operation was recommended. The clinical test results are summarized in Table 1.

### 2.2. Literature Data Investigation

We conducted a comprehensive review of recent English-language literature regarding ovarian tumors, lobular breast carcinoma, and tumor-to-tumor metastasis. Our review encompassed the utilization of the PubMed electronic database alongside other relevant sources. We used a set of predetermined keywords, including “breast cancer”, “lobular carcinoma”, “ovarian metastases”, “ovarian metastases from non-gynecologic primary sites”, and “tumor-to-tumor metastases”. We meticulously overviewed a database comprising 127 studies addressing breast and ovarian neoplasms, comprising both individual studies and reviews.

Subsequently, we curated this database, eliminating duplicate entries and articles falling outside the scope of our investigation, resulting in a refined dataset of 80 relevant papers. Publications lacking sufficiently explained results, discussing irrelevant topics, or presenting redundant information were systematically excluded from further consideration. Ultimately, 42 articles met our inclusion criteria.

We categorized the remaining articles into two distinct datasets. The first dataset comprised a study investigating breast cancer (both primary and metastatic tumors) and ovarian cancer, with a specific focus on elucidating current insights into ovarian metastases. The second dataset comprised reports documenting tumor metastasis within an ovarian tumor. The algorithm employed for the selection of articles is explained in Figure 1.

## 3. Results

### Gross and Microscopic Evaluation

The patient had undergone a bilateral salpingo-oophorectomy. Further postoperative gross and microscopic examination of the excised tissues was made. On gross examination, the resected right specimen showed a 10/8/7 cm right ovarian mass well circumscribed with a tan-yellow to brown aspect on the cut surface. Sections from this mass were harvested and investigated histopathologically. The 4 µm thick formalin-fixed paraffin-embedded tissue samples were stained with hematoxylin and eosin (HE). Additional slides from the ovarian tumor were immunohistochemically stained with AE-1/AE-3 cytokeratin, mammaglobin, GCDFP-15, inhibin, and calretinin. In addition, we used histological techniques: periodic acid–Schiff alcian blue (PAS-AA) for mucins and the Gordon–Sweets technique for reticulin fibers.

The HE slides revealed a biphasic tumor with circumscribed areas of round tumor cells bordered by variable cellular spindle cells with focal fascicular growth. The fusiform cells had bland spindle to ovoid nuclei and scant cytoplasm resembling the stroma of the ovarian cortex. The round cells were arranged in solid areas and only focally in cords or sheets. These cells were small, monomorphic, and without marked atypia, necrosis, or atypical mitoses. The stromal responses in the round cell areas lacked desmoplastic reactions (Figure 2).

The PAS-AA revealed in the round cells tumor areas of positive (blue) material in the cytoplasm of these cells but also focally in the adjacent stroma (extracellular). The reticulin highlighted the aspect of the disrupted architecture of the stroma in the round-shaped cell areas and underlined the areas with the spindle cells. The fibroma areas had a pericellular reticulin staining pattern (Figure 3).

We studied the biphasic tumor areas immunohistochemically and observed that the round cells were positive for AE-1/AE-3 cytokeratin, mammaglobin, and GCDFP-15, and the fusiform tumor cells stained for inhibin and calretinin (Figure 4 and Figure 5).

## 4. Discussion

Ovarian metastases from breast cancer account for 3–30% of all ovarian ovarian neoplasms [1,2,3,4,5,6,7,8,9,10,11]. We identified both lobular and invasive ductal breast carcinoma studies with metastasis to the ovary, but the cases presented in the current literature data were few. Brigorie V et al. identified 29 breast carcinomas with ovarian metastases without specifying the CA-125 level [1]. Pimentel C et al. presented 28 such cases with a mean value of 488 and only 9 cases with a CA-125 level higher than their reference value of 35 U/L [5]. Zhang JJ et al. reported 6 breast tumors metastatic in an ovary from 177 tumors identified, but the histological type was not specified, and the CA-125 level was higher than 35 U/L in 65% of the cases included in their study [7]. In the rest of the analyzed studies, isolated cases of ovarian metastases of breast lobular carcinoma or tumors with primary localization in other organs were reported [2,3,4,6,8,9,10,11]. The types of primary tumors metastatic to the ovary, their incidence, CA-125 levels, and the number of cases identified in the literature database are summarized in Table 2.

Invasive ductal carcinoma, not otherwise specified, stands as the predominant malignant breast tumor type. However, ILC of the breast, characterized by unpredictable behavior, is more prone to multifocal and distant metastases. The loss of the adhesion molecule E-cadherin, a hallmark of ILC, is implicated in the dissemination of tumor cells to distant sites and the occurrence of bilateralism. Metastases from breast ILCs have been documented in the bones, lungs, pleura, soft tissue, and liver [1,2,3,4,5,6,7,8,9,10,11,20,21,22,23,24,25,26,27,28,29].

The gastrointestinal tract represents a metastatic site in 6–18% of ILC cases, with the stomach being the most frequently affected site, followed by the colon and rectum [29]. Furthermore, metastases to the genital tract, including the uterus, cervix, and ovary, have been reported [25,26].

Ovarian fibromas typically present in middle-aged postmenopausal women and can simulate malignant tumors when associated with pleural effusions or ascites, posing diagnostic challenges preoperatively. Histologically, fibromas often resemble leiomyomas [34,36,42,43]. Additionally, while elevated levels of CA-125 are rare in fibromas, some studies suggest that mechanical stimulation of the peritoneum in the presence of ascites can result in elevated CA-125 levels, potentially leading to misdiagnosis of malignant ovarian tumors. Rare instances have been reported wherein fibromas spread and adhere to adjacent tissues, resulting in fatal outcomes [35]. Moreover, normal serum CA-125 levels are observed in up to 35% of cases of ovarian metastases from non-gynecological neoplasms [8].

In this study, we present a case of ILC metastasizing to an ovarian fibroma. This case posed diagnostic challenges as both breast tumors and primary ovarian tumors can coexist. In our patient, a prior diagnosis of ILC was compounded by elevated CA-125 levels and a ROMA score indicating an increased risk of ovarian cancer development.

Studies show that patients with a history of breast cancer are more predisposed to develop primary ovarian tumors [9]. The likelihood incidence ratio of a primary ovarian tumor to metastasis of a breast neoplasm is 7:1 in a 12-year retrospective study [43]. Also, less than 10% of breast cancer patients have clinical signs of distant metastasis at the moment of the initial diagnosis [9]. Women with hereditary conditions and BRCA-1 and BRCA-2 mutations have a higher risk of developing both primary ovarian and breast tumors [4]. Also, there are therapeutic implications. The primary ovarian cancer treatment is surgical debulking, but the optimal treatment for ovarian metastasis from other primary sites has not been established [7]. In the analyzed case, the morphological characteristics of the tumor cells within the areas of metastatic involvement closely resembled the histopathological profile of lobular carcinoma of the breast, exhibiting pathognomonic features identifiable through standard staining techniques. Various staining techniques, such as hematoxylin and eosin and PAS-AA [44], are commonly employed for diagnosing metastasis to the ovaries, aiding in the accurate identification of abnormal cells. To ascertain the cellular lineage and primary tumor site within the mammary gland, immunolabeling techniques were employed. Positive immunoreactivity for cytokeratin confirmed the epithelial origin of the tumor, while the presence of GCDPF-15 and mammaglobin indicated the breast as the primary tumor site. Furthermore, to distinguish the fibroma from other similar sex-cord stromal tumors, staining for reticulin fibers was utilized. In fibromas, a pericellular reticulin staining pattern is typically observed, contrasting with the varied patterns observed in tumors such as diffuse adult granulosa cell tumors, smooth muscle tumors, or mesenchymal tumors [45]. Disruption of reticulin fiber architecture, similar to that observed in areas of metastatic lobular carcinoma, is indicative of malignancy.

During immunohistochemistry, a diagnostic technique that may use several specific antibodies to detect proteins [46], the fibroma exhibited positive staining for inhibin and calretinin. Recent investigations have proposed the utility of inhibin and calretinin in diagnosing ovarian sex-cord stromal tumors, with fibromas demonstrating positivity for these markers in approximately 25–30% of cases studied [37,47,48].

The morphological characteristics of the metastatic lobular breast carcinoma were further elucidated through PAS-AA staining. Notably, the tumor cells exhibited blue staining of intra- and extracellular material suggestive of mucin production. While intracellular and extracellular mucin secretion is a recognized feature of lobular breast carcinoma, the identification of such mucins within the metastatic tumor is a rare occurrence [30,31,33,49]. Some studies suggest that mucin production may signify aggressive behavior in lobular carcinoma [31]. In the case under consideration, mucinous material was predominantly localized within the cytoplasm of tumor cells, with focal extracellular distribution.

Another unique aspect in our case is tumor-to-tumor metastasis (TTM). Our second dataset included reports of tumor metastasis in an ovarian tumor. TTMs turned out to be a rare finding. By consulting the English-language literature data, we have identified only eight TTMs, with one case of ILC metastatic to a sex-cord stromal tumor. Strobel et al. described a triple-negative ILC metastasis to an ovarian fibrothecoma [19]. Cases of follicular lymphoma, appendiceal adenocarcinoma, squamous-cell carcinoma, and breast ILC metastatic to ovarian mature teratoma have been described [12,14,50].

## 5. Conclusions

We report a rare case of tumor-to-tumor metastasis of an uncommon mucin-producing ILC. To the best of our knowledge, this is the first case of TTM due to lobular carcinoma of the breast to an ovarian fibroma. Awareness of this phenomenon is important for the therapeutic management of the patient. If for primary ovarian masses, the treatment is surgical, for TTMs, the therapeutic approach should be made by a multidisciplinary medical team.

## Figures and Tables

**Figure 1 diagnostics-14-00953-f001:**
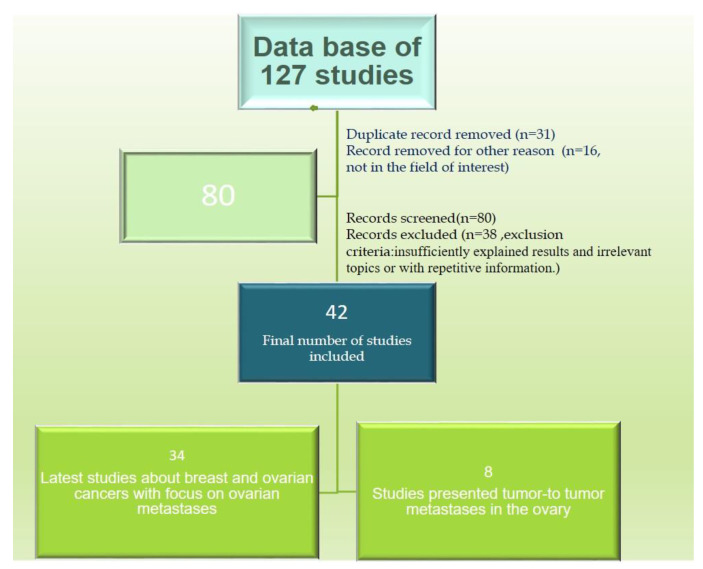
Algorithm used for article selection; n = number of articles.

**Figure 2 diagnostics-14-00953-f002:**
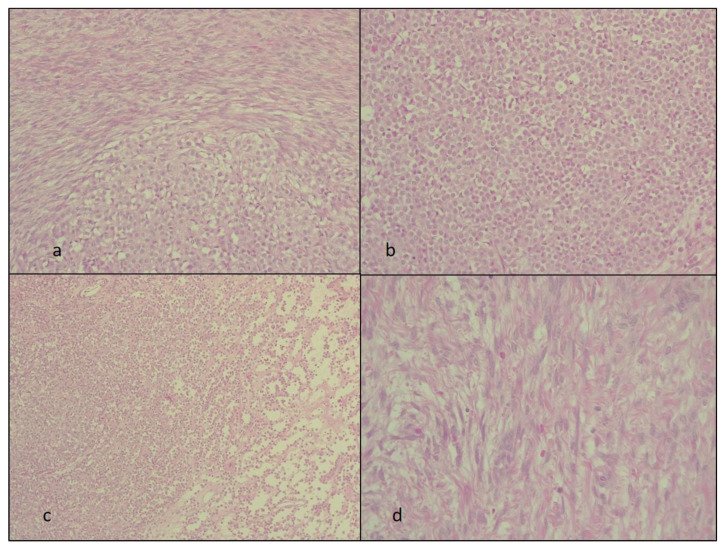
The hematoxylin and eosin staining. The HE slides with the biphasic tumor showing (**a**) the tumor-in-tumor aspect objective (at ×40 Ob.), (**b**,**c**) the ILC metastasis in detail at ×20 and ×40 Ob., respectively, and (**d**) the spindle cells fibroma area (at ×40 Ob.). Ob = the microscope objective.

**Figure 3 diagnostics-14-00953-f003:**
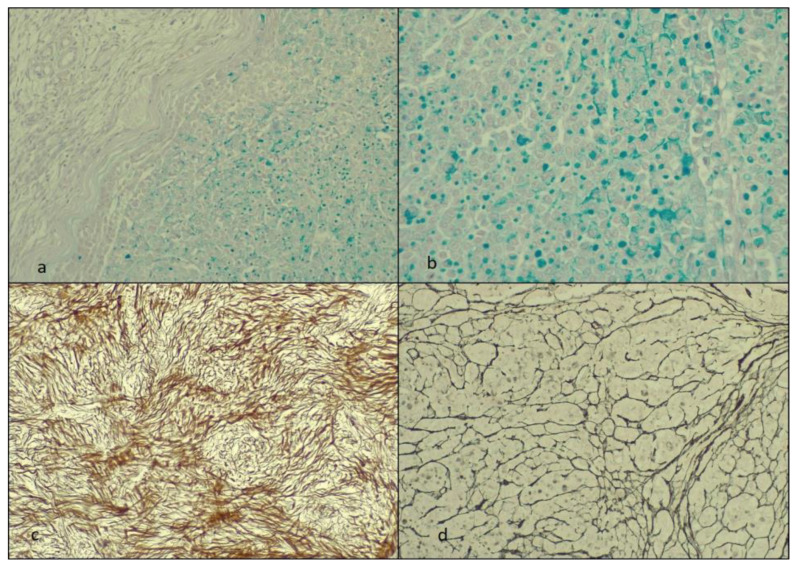
The staining with PAS-AA and reticulin: (**a**,**b**) the PAS-AA reaction at ×20 and ×40 Ob., respectively; (**c**,**d**) reticulin stain (at ×20 and ×40 Ob., respectively) with (**c**) the fibroma areas and (**d**) the ILC.

**Figure 4 diagnostics-14-00953-f004:**
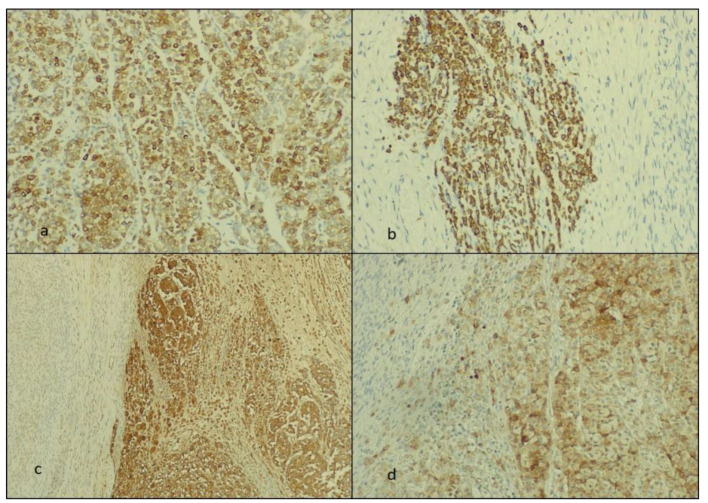
The immunohistochemically staining: (**a**,**b**) the AE-1/AE-3-positive ILC metastasis at ×40 and ×20 Ob., respectively; (**c**) the mammaglobin-positive cells of the ILC and (**d**) GCDFP-15-positive ILC areas at ×20 and ×40 Ob., respectively.

**Figure 5 diagnostics-14-00953-f005:**
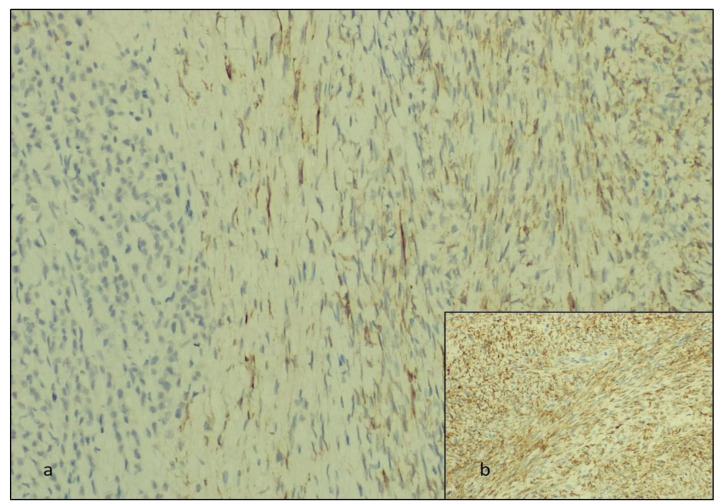
The immunohistochemical staining. The fibroma areas are positive for (**a**) inhibin and (**b**) calretinin (at ×40 Ob.), respectively.

**Table 1 diagnostics-14-00953-t001:** Tumor markers values.

Marker	Results	Reference Value (Postmenopausal) [39,40,41]
CA-125	751 U/L	<46 U/L
HE-4	381 pmol/L	<140 pmol/L
ROMA Score	95%	>2.99 (high risk for ovarian cancer)

CA-125 = cancer antigen 125, HE-4 = human epididymis protein, ROMA score = risk of ovarian malignancy algorithm.

**Table 2 diagnostics-14-00953-t002:** Summary of the studies reporting ovarian metastasis in the English-language literature.

References	Type of Study	Ovarian Metastases: Primary Sites (Histopathological Diagnosis of the Primary Tumor and/or Number of Cases)	CA-125 Levels	Number of Cases
Bigorie, V.; et al. [1]	Retrospective	Breast (16 cases with invasive ductal carcinoma (IDC), 12 ILC and 1 unknown)	NS	29
Akizava, Y.; et al. [2]	Case report	Breast (bilateral carcinoma ILC and IDC)	692 U/mL	1
Yamaguchi, T.; et al. [3]	Case report	Gastrointestinal tract (gastrointestinal stromal tumor)	818 U/L	1
Makris, G.M.; et al. [4]	Case report	Brest (ILC)	78 UI/L	1
Pimentel, C.; et al. [5]	Retrospective	Breast cancer (17 ILC, 7 IDC, 2 mixed ILC and IDC), 2 different histology	488 IU/L mean value (9 cases with >35)	28
Wong, Y.M.; et al. [6]	Retrospective	Breast (ILC)	NS	NS
Zhang, J.J.; et al. [7]	Retrospective	Colorectal (68), stomach (61), appendix (12), biliary tract (7), pancreas (7), breast (6), small intestine (5), lung (3), bladder (2), unknown sites (6)	>35 U/L in 65% of the patients	177
Kemps, G.P.; et al. [8]	Case report	Colon (adenocarcinoma not otherwise specified)	18 kU/L	1
Porfirys, O.; et al. [9]	Case report	Kidney (clear-cell renal cell carcinoma)	103.3 U/mL	1
Li, Z.; et al. [10]	Retrospective	Pancreas (5 cases with pancreatic ductal adenocarcinoma, 2 cases with pancreatic cystadenocarcinoma)	2 patients with elevated values -NS	7
Wang, S.D.; et al. [11]	Case report	Pancreas (pancreas mucinous adenocarcinoma)	412 U/L	NS

ILC = invasive lobular carcinoma, IDC = invasive ductal carcinoma, NS = not specified.

## Data Availability

The data presented in this study are not publicly available due to privacy.

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
