# Peer review of "Mucin-Producing Lobular Breast Carcinoma Metastasis to an Ovarian Fibroma: Histopathological and Immunohistochemical Analysis of a Rare Case and Literature Review"

_diagnostics, 2024, doi:10.3390/diagnostics14090953_

Round 1

Reviewer 1 Report

Comments and Suggestions for Authors

Very interesting case presentation and review of literature,

Would be a valuable addition to current literature as a metastatic breast carcinoma and a primary ovarian fibroma is very rare.

English language in the text needs revision and review by a native English speaker.

Comments on the Quality of English Language

Low quality English language, needs major revision.

Author Response

For the 1. Review Report (Round 1). Thank for reviewing our paper. We have improved the English language as you requested. The authors.

Reviewer 2 Report

Comments and Suggestions for Authors

At line 44-45 ‘Invasive lobular carcinoma (ILC) usual 44 sites of metastases include the bones, liver and lymph nodes’ should be checked.

At line 54-55 ‘Fibromas are common benign stromal ovarian sex-cord stromal tumors diagnosed 54 representing 4% of all ovarian tumors’ should be checked … diagnosed is not necessary 

At line 57-59 sentence should be rewritten  

There are some sentences that have English language writing problem.   

Paragraph regarding literature database for systematic review (in discussion) should be written in section of material method and also figure 5 should be taken to material method.

In table 2, It has been reported that the incidence of ovarian metastases from breast cancer was 3-30% according to the literature belong Bigorie V et all 2010. But there is no any result for metastatic ovarian cancer incidence of this study. It should be checked 

Also some case reports in this table have incidence results. It does not make sense that a case report provides an incidence result. 

At line 170-171 ‘On ultrasound examination, fibromas show similar- 170 ities with leiomyomas, and the aspects of ascites fluid’ In this sentence, what is the mean of the aspects of ascites fluid. Should be checked 

At line 183-184, ‘Studies show that patients with history of breast cancer are 3 to 7 times more predisposed to develop primary ovarian tumors. This data need literature.

Comments on the Quality of English Language

English language writing should be corrected. There are many language problems in this manuscript. 

Author Response

For the 2. Review Report (Round 1). Thank for reviewing our paper. We have improved the English language, added new references and rewritten the sentences as you requested. We have changed also the material and methods as you suggested by adding the information and the figure from the discussion section. We have also checked the table 2. We have removed the column with the incidence. We felt that if you were confused by the way, it was set and explained this incidence in the table other will be. So we have shortened the table to be more concised. All the paragraphs were verified. The information in the specified lines were changed according to your suggestions. Thank you for your ideas and your time taken to review this manuscript. The authors.

Round 2

Reviewer 1 Report

Comments and Suggestions for Authors

The revised manuscript is suitable for publication.

Reviewer 2 Report

Comments and Suggestions for Authors

there is no any sugestion , thank you for your effort.